# Sketching based Representations for Robust Image Classification with Provable Guarantees

**Nishanth Dikkala**[*]
Google Research
nishanthd@google.com

**Sankeerth Rao Karingula**[*]
Google Research
sankeerth1729@gmail.com

**Raghu Meka**[*]
UC Los Angeles
raghuvardhan@gmail.com

**Jelani Nelson**[*]
UC Berkeley
minilek@gmail.com

**Rina Panigrahy**[*]
Google Research
rinap@google.com

**Xin Wang**[*]
Google Research
wanxin@google.com

## Abstract

How do we provably represent images succinctly so that their essential latent attributes are correctly captured by the representation to as high level of detail as possible? While today's deep networks (such as CNNs) produce image embeddings they do not have any provable properties and seem to work in mysterious non-interpretable ways. In this work we theoretically study synthetic images that are composed of a union or intersection of several mathematically specified shapes using thresholded polynomial functions (for e.g. ellipses, rectangles). We show how to produce a succinct sketch of such an image so that the sketch "smoothly" maps to the latent-coefficients producing the different shapes in the image. We prove several important properties such as: easy reconstruction of the image from the sketch, similarity preservation (similar shapes produce similar sketches), being able to index sketches so that other similar images and parts of other images can be retrieved, being able to store the sketches into a dictionary of concepts and shapes so parts of the same or different images that refer to the same shape can point to the same entry in this dictionary of common shape attributes.

## 1 Introduction

A first step in many computer vision systems is to learn a succinct representation of an input image (or a set of images) which retains all the *useful* information about the image. Today, there are several deep networks that can take images and produce useful embedding representations [4, 7, 2]. Deep approaches are extremely complex and hard to understand. Moreover, for the tasks they aim to solve, it is unclear if we need such a non-interpretable ([6, 1]) approach which is also known to be very brittle ([17]). Previously hand-crafted feature construction approaches such as SIFT [13] used to dominate. These were also heuristic and not theoretically guaranteed to work. In addition, they focused heavily on local features which might not be good at understanding important global information such as shape.

Ideally we would like representations of images to be such that they capture the essential latent semantic attributes of the image. If the images are produced by some generative process then the latent attributes correspond to parameters used in the generative process. Further, given a large collection of images, it can be desirable to create a dictionary or an index of commonly occurring latent attributes so that each image may be viewed as a set of pointers into this global dictionary. To study this problem theoretically, we use a simple generative process for producing images using

---

[*]Alphabetical author order.

36th Conference on Neural Information Processing Systems (NeurIPS 2022).

polynomial functions. By taking thresholded versions of polynomial functions one can generate simple shapes such as circles, ellipses, hyperbolas and by taking intersections of such functions one can generate polygons such as rectangle, triangles etc. By taking a union of such intersections one can generate fairly complex images – in fact, much of computer graphics is generated using such principles.

Since this process is fully formalized, it is now feasible to study questions such as: *Do deep networks learn representations that are isomorphic to the parameters that generated the image?* While today's deep networks do not provably have such properties about being able to recover the polynomial coefficients used in an image, we provide a method that can produce a sketch that is provably equivalent to the parameters that generated the image. While our sketch is not identical to the polynomial coefficients, we show that it is equivalent in a smooth manner – that is there exist smooth functions mapping one to the other and vice versa. Further our sketches can be easily used to reconstruct the original image which means that our representations are robust across different types of shapes.

The main idea behind our sketching algorithm is fairly simple and uses a locality sensitive hash (LSH) table. We partition the pixels of the image into regions based on the LSH and train specific parameters within each hash bucket so that those parameters can be used to reconstruct that region of the image. By using a few such LSH tables and training parameters in those buckets we obtain a succinct sketch that is able to provably reconstruct the image. This provides several advantages: 1) our sketches are invertible, that is one can reconstruct the original image; 2) they are semantically meaningful since they are isomorphic (using a smooth transform) to the latent parameters. This in turn means that any classification of shapes that can be done using the latent parameters can also be done using LSH sketches. If we think of each shape type (such as ellipse) as a separate region in coefficient space that is well separated from other regions by a polynomial threshold function (PTF) over the coefficient vector, then by training such a PTF classifier over the sketches of such images we obtain a robust classifier for such shapes. In fact in the coefficient space (which is homeomorphic to the set of produced sketches), we can view each shape type itself as an intersection of PTFs and thus the above tools for representing shapes in images could be used for representing shape-types in sketch space.

Our sketches also have the nice property that the sketch of a part of the image can be easily obtained as some of the dimensions in the full sketch. This means that one can create a secondary LSH index of sketches for different parts and different shapes in the image over a large collection of images. This gives a global dictionary of shape attributes so that each image merely becomes a set of points into this global dictionary. If we think of each shape-type as a disjoint compact region in coefficient space, then each contiguous set of buckets in this secondary index forms a shape-type and thus this secondary index can be viewed as a dictionary of shape-types.

We show the following results. An image can be viewed as map from pixel location to pixel values.

**Proposition 1.1** (Informal, Image sketches and their properties)**.** *1. **Low degree polynomials**: Given a grayscale image where the map from pixel position to the pixel value is a degree $s$ polynomial, we construct sketch vector of size $O(s)$ that is able to reconstruct the image (see Theorem 4.4). Further the coefficient vector of the polynomial can be recovered by applying a linear transform over the sketch vector (see Appendix).*

2. ***Polynomial threshold functions**: If the image is a PTF where the underlying polynomial is of degree $s$ we can get a sketch vector of size $O(s)$ that can reconstruct the image (Theorem 5.4).*

3. ***Parametric isomorphism**: The sketch is "isomorphic" to the original parameters used in producing the image. Furthermore, the transform from the coefficient parameters to the sketch is "smooth" (continuous and differentiable) (Lemma 5.5); whereas for PTFs the image itself (or a random projection of it) is not differentiable in the underlying coefficients.*

4. ***Union/Intersections of PTFs**: A union of intersection of $k$ PTFs each of degree at most $s$ can be converted into a sketch of size $poly(ks)$ from which the image can be reconstructed. (claim 6.2)*

5. ***Local property**: By projecting to specific dimensions, for any patch in the image one can obtain a sub-sketch from the sketch corresponding to that patch. One can create an index of*

*patch sketches that can be used to retrieve patches from other images that are similar to a given patch (claim 6.3).*

**Theorem 1.2** (Learning Shape concepts, see claims 7.3, 7.4)**.** *Assuming shapes (such as ellipse) can be expressed as a region in coefficient space, and that each such region can be expressed as a PTF over the coefficient vector, and that these regions are well separated, our sketching methods can be used to build a robust high margin classifier for shapes that identifies these regions in sketch space and classifies an image sketch to its shape.*

## 1.1 Related Work

There is a tremendous amount of work on building features specifically for learning images similar to the ones we consider in the computer vision literature (for instance, *SIFT* features [13]). The literature is too vast to survey here — see [18] and references therein for instance. However, by and large while these methods are empirically well-understood they do not come with rigorous guarantees in terms of compressing images.

Learning good representations has also been posited as a key reason for the success of deep neural networks [2]. Deep representations of complex inputs such as text and images are often used to compare the underlying objects and transfer to new classification problems [20, 16]. However there is little theoretical understanding of the neural representations computed by such networks.

Locality sensitive hashing was originally introduced by [8] in the context of speeding up nearest-neighbor computations but has since found many applications in sketching in the streaming community. The LSH framework has been applied to deep learning and vision settings as well but our approach is tangential to these. In particular, it has mainly been used to speed up computations of various neural network architectures (see e.g., [9, 3]). We also study a more general paradigm recursive LSH based sketching. We also refer the reader to [19] for a survey of LSH.

There is also extensive work on learning PTFs and intersections of PTFs (e.g., [12, 10, 11]) in the high-dimensional setting under distributional assumptions. We also have distributional assumptions however the focus is on the very low-dimensional setting and dealing with an unsupervised learning problem with emphasis on computing compact representations.

## 2 Setup and Preliminaries

We focus on inputs which are 2D grayscale images which are square in shape for simplicity. All our algorithms and results easily extend to non-grayscale rectangular images as well. We view an image $I$ as a real-valued function on a continuous 2-D domain $I : [0,1]^2 \to [0,1]$ mapping positions within a rectangular window to grayscale values.

**Sampling Pixels of an Image:** Given an image, we will assume the learner has $n$ evaluations of $I$ at points within $[0,1]^2$ sampled uniformly at random. That is we are given $(x_1^i, x_2^i)_{i=1}^n \sim U[0,1]^2$. These points can be thought of as pixels in a digital representation of the image. The vector of pixel values at the sampled positions is denoted by $\mathbf{v} \in [0,1]^n$. We denote these $n$ points as samples of $I$.

**Underlying Generative Process for Images:** We will model images as being the outputs of a generative process. We will consider the following simple generative processes.

**Definition 2.1** (Polynomial Images)**.** The map $I : [0,1]^2 \to [0,1]$ is a degree-$s$ polynomial in 2 variables $(x_1, x_2)$.

**Definition 2.2** (Thresholded Polynomial Images)**.** Given a polynomial $P(x)$, a polynomial threshold function (PTF) $f_P : \mathbb{R}^d \to \{0,1\}$ is defined as $f_P(x) = \mathbb{1}\{P(x) \geq 0\}$. A thresholded polynomial image is a PTF on $[0,1]^2$.

Rather than bounding the degree explicitly, we sometimes impose a 'smoothness' constraint on the polynomial generating our image.

**Definition 2.3** ($\gamma$-Smoothness)**.** We say a function $f : [0,1]^2 \to [0,1]$ is $\gamma$-smooth if $\sum_{k,l=0}^\infty c_{k,l}^2 (k^2 + 1)(l^2 + 1) \leq \gamma^2$ where $c_{k,l}$ are the coefficients of the Fourier series of $f$ restricted to $[0,1]^2$.

We will also talk about classes of images which together can be thought of as constituting a *concept*. We assume that a class of images forms a contiguous region in the underlying parameter space which generates our image.

**Locality Sensitive Hashing**   We will construct our image sketches using a form of hashing which is sensitive to distance called locality sensitive hashing which is a form of hashing defined on a metric space where nearby points hash to same buckets and far away points hash to different buckets.

Originally introduced for solving the approximate of exact Nearest Neighbor Search problem, LSH has since found applications in many other problem areas. Next we define the specific LSH family we use.

**Definition 2.4** (Hyperplane LSH Family for Euclidean Distance)**.** The hyperplane LSH family $\mathcal{H}_{\|\|_2}(k, w)$ for the metric space $(\mathbb{R}^d, \|.\|_2)$ is one where each hash function is defined by a set of hyperplane inequalities (we use axis-aligned hyperplanes in some settings). We sample $k$ random directions $a_1, \ldots, a_k$ where each coordinate of each $a_i$ is an independent normal variable. In addition we sample $b_i \sim Unif[0, \epsilon]$ independently for $i \in [k]$. Given a hash function $h \in \mathcal{H}_{\|\|_2}(k, w)$ defined by the hyperplane directions $\{a_i\}_{i=1}^k$ and shifts $\{b_i\}_{i=1}^k$, $h(x) = \left( \left\lfloor \frac{a_i^\top x + b_i}{\epsilon} \right\rfloor \right)_{i=1}^D$.

# 3   Representation Learning using LSH

A natural assumption about images is that how quickly the pixel-values change with respect to distance in pixel locations conveys important information for most image-processing tasks. Convolutions crucially exploit this by using the same filter across various patches. We propose to exploit this by using LSH.

**Definition 3.1.  Simple LSH-Sketch of an Image $I$.** Given $n$ uniformly random samples of $I$, we define the Simple LSH-Sketch $S(I; m, k, w)$ of an image $I$ as follows.

1. Using $m$ random functions $\{f_j\}_{j=1}^m \in \mathcal{H}_{\|\|_2}(k, w)$, map the $n$ pixel positions to corresponding hash buckets.

2. For each hash function $f_j$, we allow each hash bucket to store a scalar value. We denote the vector of the resulting bucket values by $\mathbf{z}_j \in \mathbb{R}^K$ where $K$ is the number of hash buckets. The learnt value in each bucket will act as an approximation to the image in the region covered by that bucket.

Effectively, the Simple LSH sketch is a piecewise regression model where the pieces are determined by partitions induced by LSH. It can also be considered as a form of a smoothening of the image $I$ (analogous to Gaussian smoothening).

**Procedure for Reconstructing an Image from its Simple LSH-Sketch**   Given a Simple LSH-Sketch $S(I; m, k, w)$, we can reconstruct an approximation of $I$ as follows. For any position $(x_1, x_2)$, the reconstructed pixel value is defined as the average of the values stored in all the buckets that the point $(x_1, x_2)$ hashes to under the $m$ hash functions constituting the sketch. Let $K$ denote the total number of buckets per each hash function. For the pixel values at the train locations $\mathbf{v} \in [0, 1]^n$, since the reconstruction is a linear operation it can also be represented in matrix form as $\mathbf{u} = A^\top \mathbf{z}$, where $\mathbf{z} = (\mathbf{z}_1, \mathbf{z}_2, \ldots, \mathbf{z}_m) \in \mathbb{R}^{mK}$ is the vector of all bucket values concatenated and $A \in \mathbb{R}^{mk \times n}$ is a matrix mapping pixels to buckets such that

$$A(i, j) = \frac{1}{m} \text{ if } (x_1^j, x_2^j) \text{ maps to bucket } i, \quad A(i, j) = 0 \text{ otherwise.}$$

**Computing the Optimal Bucket Values**   Given the above procedure to reconstruct the image, we now define the learning problem which we solve to learn the bucket values $\mathbf{z}_j$ for each hash function in the Simple LSH-Sketch. The values stored in the buckets aim to minimize the reconstruction error between the predicted pixel values $\mathbf{u}$ and the true pixel values $\mathbf{v}$. Our goal is to minimize the reconstruction error on a randomly drawn test pixel.

$$\mathcal{L}(\mathbf{u}, I) = \mathop{\mathbb{E}}_{\mathbf{x} \sim U[0,1]^2} [\ell(u(\mathbf{x}), I(\mathbf{x}))], \tag{1}$$

where $\ell(u, v) = (u - v)^2$ for polynomial images and $\ell(u, v) = \max(0, 1 - uv/\epsilon)$ (hinge loss) for PTF images. To achieve this goal, we minimize the $\ell_2$-regularized training loss $\mathcal{L}_n(\mathbf{u}, I) = \sum_{i=1}^n \frac{1}{n} \ell(u_i, v_i) + \lambda \|\mathbf{z}\|_2^2$.

## 3.1 Multi-Level LSH Sketch (LSH-DAG Sketch)

The Simple LSH-Sketch works well for simple images defined using polynomials. For more complex images, it helps to build a multi-level LSH-Sketch defined using an underlying directed acyclic graph (DAG). We call this the LSH-DAG sketch.

**Definition 3.2** (LSH-DAG Sketch). A function class $F = \{f : [n]^2 \to \mathbb{R}\}$. is a DAG where for every internal node the outgoing edges are labeled according to the indices of buckets of a function from a LSH family. In addition, the nodes with no outgoing edges (i.e., leaves) are labeled with functions $f_\ell : [0, 1]^2 \to [0, 1]$ that are obtained as the threshold of a degree $k$ polynomial on the input.

**Recursive LSH sketching algorithm:** Algorithm 1 describes how to compute a good multi-level LSH-Tree sketch of an image. At a high level, it computes an LSH-Tree where regions that are hard to learn at a certain resolution are recursively expanded into smaller regions to get a Tree of sketches.

---

**Algorithm 1** Recursive Sketching Subroutine. Image or the region in a bucket may optionally be centred before sketching

---

1: Given a function $f : [0, 1]^2 \to [0, 1]$ pick a resolution $r$ and map the pixels to LSH buckets with constant bucket width parameter $r$
2: Train a sketch in the buckets to best predict the pixel values in the image restricted to the region mapping to that bucket
3: Mark a bucket as invalid if the prediction error for the pixels mapping to the bucket is not negligible
4: Recursively compute a finer LSH sketch if the bucket is marked invalid by dividing the region into smaller regions with resolution $r/2$ (do not expand buckets whose region is fully covered by other valid buckets).

---

**Merging near duplicate sketches in the LSH-Tree to get an LSH-DAG:** we can merge sub-sketches and even entire sub-trees that are near duplicates to get a DAG structure instead of a recursive tree by using secondary LSH table of sketches. **Inputs**: A bucket $B$ of a recursive LSH sketch, A global secondary LSH table of sketches $H_{sketches}$.

---

**Algorithm 2** Recursive merging into LSH table of sketches

---

1: If the bucket $B$ is a marked as a valid (thus leaf level bucket that is not further expanded) then store the sketch in the bucket into LSH table $H_{sketches}$. And replace the bucket contents by a pointer to the index of the entry in $H_{sketches}$
2: If a bucket $B$ is not a leaf level bucket (so it is expanded into further buckets) then recursively merge the sketches in each of the component buckets. Look at the vector of merged sketches in the component buckets and hash to store this vector into $H_{sketches}$ and store only the index in $B$

---

# 4 Warm-up: Sketching Low-Complexity Polynomials

Polynomial images are simpler to sketch. For instance, a random projection of the image of a degree-$s$ polynomial to $O(s^2)$ dimensions gives all 4 desiderata, (i) reconstruction, (ii) small sketch, (iii) recoverability of parameters, (iv) smoothness as a function of polynomial coefficients. However, it will be beneficial to see how we can use LSH to sketch polynomial images as the analysis will help us extend to more complex images in later sections. In fact, we will show that LSH works not only for sketching degree-bounded polynomials but for polynomials of arbitrary degree as long as their Fourier spectra are appropriately bounded as described in Assumption 4.1.

**Assumption 4.1.** We assume that the polynomial $p(x_1, x_2)$ generating the image satisfies:

1. Suppose $p(x_1, x_2) = \sum_{k,l=0}^{\infty} c_{k,l} \exp(2\pi i k x_1) \exp(2\pi i l x_2)$ is the Fourier series of $p$. We assume that $\sum_{k,l=0}^{\infty} c_{k,l}^2 (k^2 + 1)(l^2 + 1) \leq \gamma^2$ for some constant $\gamma$.

2. In the region $[0, 1]^2$, the Lipschitz constant of $p(x_1, x_2)$ is bounded by $\Lambda$.

Note that our assumption on the Fourier spectrum of $p$ is satisfied by bounded polynomials with bounded degrees in particular.

We start by first studying the limiting behavior of the axis-aligned LSH-Sketch as the number of hash functions $m \to \infty$.

**Definition 4.2** (Infinite LSH-Sketch). Given an image $I$, we sample $m$ hash functions from $\mathcal{H}_\perp(m, w_1, w_2)$ for $w_1, w_2 \sim U[1/4, 1/2]$ and we analyze the properties of the resulting sketch as $m \to \infty$.

In this scenario, the metric we focus on for reconstruction is the mean squared error (MSE). Given the LSH Sketch $S(I; m, w_1, w_2)$, let $K \leq \lceil 1/w_1 \rceil \times \lceil 1/w_2 \rceil$ denote the total number of buckets per hash function. Let $A \in \mathbb{R}^{mK \times n}$ denote the mapping from the $n$ sampled pixel positions $\{(x_1^i, x_2^i)\}$ to buckets. We minimize $\frac{1}{n} \|A^\top \mathbf{z} - \mathbf{v}\|_2^2 + \frac{\lambda}{n} \|\mathbf{z}\|_2^2$.

We can obtain a closed form expression for the $\hat{\mathbf{z}}$ which minimizes the above objective. $\hat{\mathbf{z}} = A(A^\top A + \lambda I_n)^{-1} \mathbf{v}$.

Next, we observe that $\lim_{m \to \infty} A^\top A$ is a shift-invariant smoothening operator (convolution matrix) which attenuates higher frequencies in the image signal.

**Lemma 4.3.** $\lim_{m \to \infty} A^\top A$ *is a shift-invariant (convolution) matrix which acts also as a smoothening matrix. In the Fourier space, convolution of a signal $I$ with $\lim_{m \to \infty} A^\top A$, corresponds to pointwise multiplication of the Fourier spectrum of $I$ given by $\tilde{I}(k, l)$ with $\mathrm{sinc}(w_1 k)^2 \, \mathrm{sinc}(w_2 l^2)$.*

This realization from Lemma 4.3 suggests a Fourier style analysis of the solution $\hat{\mathbf{z}}$. Due to Assumption B.1 as well we can focus on the first few Fourier coefficients of the image and ignore the rest. This enables us to show the following.

**Theorem 4.4** (Informal). *With probability $\geq$ 0.9, for $n = \Theta(\gamma \Lambda / \mathrm{poly}(\epsilon))$ samples, $\mathbb{E}_{\boldsymbol{x} \sim U[0,1]^2} \left[ (u(\boldsymbol{x}) - I(\boldsymbol{x}))^2 \right] \leq \tilde{O}(\epsilon)$.*

In addition to Theorem 4.4 we also show that the resulting sketch has continuous and bounded derivatives with respect to the $\boldsymbol{\theta}$ (Lemma B.13) and that there exists a linear transformation $R$ such that $R\hat{\mathbf{z}} = \boldsymbol{\theta}$ with high probability (Lemma B.14).

Finally, we show how we can obtain a sketch with a finite $m$ which approximately still satisfies the above properties.

*Remark* 4.5. [sketches are resilient to sampled pixel positions] The above methods let us produce sketches of smooth images from sampling a few pixels in the image. The sketches only need random sampling and we can compare images even when different random pixel positions are sampled.

## 5 Sketching Thresholded Polynomial Images

Similar to the polynomials setting, we assume that the Fourier spectrum of the underlying polynomial $p$ which we threshold is well-behaved. In addition the previous assumptions we had in Section 4 we also assume an additional condition about the boundary of the PTF. First we present a definition.

**Definition 5.1** ($\epsilon$-Boundary of a PTF). The region where the absolute value of the underlying polynomial is $\leq \epsilon$.

The assumptions we make about our PTF image are given next.

**Assumption 5.2.**     1. Satisfy Assumptions 4.1.

2. The total probability mass of the $\epsilon$-boundary of $p$ is at most $O(\epsilon)$. For example, this holds for an ellipse represented by $ax^2 + by^2 - 1$ when $\sqrt{a^2 + b^2} > c$ for a constant $c$.

We see how the sketch we construct can accurately reconstruct a PTF image. Due to the binary nature of the true pixel values, an objective such as cross-entropy or the hinge loss is found to

be more appropriate for computing the optimal bucket values. In particular, we use the following parametrization of hinge loss. For $y \in \{0, 1\}$, and for $\hat{y} \in \mathbb{R}$, $\ell_\epsilon(\hat{y}, y) = \max\left(0, 1 - \frac{\hat{y}y}{\epsilon}\right)$.

Similar to Section 4, we show that the sketch we get here has a low reconstruction error. In particular, we will only make errors near the boundary of the PTF and we can control this error by increasing the number of buckets per each hash function. In addition, we show that the manifold containing the sketches is Lipschitz and has bounded derivatives with respect to the coefficients of $p$.

### 5.1 Properties of The Infinite Sketch

As in Section 4, we begin by analyzing the sketch as $m \to \infty$. We first observe that as $m \to \infty$, our reconstruction procedure allows us the flexibility to express arbitrary smooth functions over a finite numbe of pixels.

**Lemma 5.3.** *As $m \to \infty$, any function $f$ which is Lipschitz and has bounded derivatives can be expressed as $A^\top z$ where $z$ are the bucket values we learn.*

Lemma 5.3 allows us to argue that we can get a small reconstruction error for PTF images.

**Theorem 5.4** (Informal)**.** *Constructing infinite sketch such that each hash function uses $O(1/\epsilon)$ buckets with $n = \Theta(\gamma\Lambda/\operatorname{poly}(\epsilon))$ samples will yield a solution with expected $0/1$-reconstruction error at most $\tilde{O}(\epsilon)$.*

A short overview of the proof of Theorem 5.4 is as follows. Recall that the hinge loss we optimize is $\max(0, 1 - uv/\epsilon)$ where $v \in \{0, 1\}$ is the true pixel value and $u$ is the predicted reconstruction value. Note that this loss is 0 when $uv \geq \epsilon$. Given infinitely many hash functions, we can realize a reconstruction such that the reconstructed value at a pixel $u$ is equal to the underlying polynomial value at $v$. This reconstruction will set $|uv| > \epsilon$ for $v$ outside the $\epsilon$-boundary and hence incur 0 hinge loss there. With our assumption on the boundedness of $\epsilon$-boundary of $p$, we get that this solution incurs a train reconstruction loss of $O(\epsilon)$ implying that the optimal solution also incurs a loss of $O(\epsilon)$. The last step is a generalization bound to ensure that this behavior approximately holds for a randomly sampled test point as well.

### 5.2 Smoothness

We show that the sketch we compute is an analytically smooth function in terms of the latent parameters used to generate the image. Note that this property is not true if we just think of the mapping to the image itself. That is, say we have a degree $d$ polynomial specified by coefficient parameters $\boldsymbol{\theta}$. Let $\mathbf{v}_{\boldsymbol{\theta}}$ denote the image generated by the PTF represented by coefficients $\boldsymbol{\theta}$. The mapping $\boldsymbol{\theta} \to \mathbf{v}_{\boldsymbol{\theta}}$ is non-differentiable whereas the mapping $\boldsymbol{\theta} \to S(\mathbf{v}_{\boldsymbol{\theta}}; m, \mathcal{H})$ is an analytic function with derivatives bounded independent of the number of pixels. This is a desirable property as it implies that any nice space of images whose latent parameters form a low-dimensional manifold will also be mapped to a low-dimensional manifold in the sketch space. Note that this not true for the image mapping as it preserves continuity but not differentiability.

**Lemma 5.5** (Sketch-Smoothness)**.** *The infinite sketch produced is smooth in the PTF parameters and vice versa. If we use the image directly as the sketch then it is not smooth (first derivative is not defined).*

The main idea behind the proof is that as the sketch is the $\arg\min$ of a convex optimization problem, we can compute derivatives of the $\arg\min$ with respect to the parameters of the objective function using the *implicit function theorem*. See C.0.2 for the details of the proof.

As in Section 4, we can again go from the infinite sketch to a finite sized sketch with both a small reconstruction error and (approximate) smoothness. Details are given in the supplementary material.

## 6 Intersection of PTFs

**Learning a rectangle** We will now show how to construct a recursive sketch of a (constant-sized) rectangle image via our algorithm 1. We will first focus only on reconstruction. Instead of keeping a scalar in each bucket, we will keep a sketch vector within each bucket that can be used to reconstruct

the region that maps to that bucket. For simplicity, let us say we use an LSH that does say a square tiling with a random offset (shift) where each bucket is a square of side length say $1/c$. This would give about $c^2$ buckets. Within each bucket we run a PTF learning algorithm giving a $O(d)$ sized sketch that can learn any edge that can be expressed as a PTF of degree $d$ as long as the tile intersects purely with at most one side of the rectangle.

Any tile-sketch that does not predict its content with high accuracy is marked as invalid and we recurse further by dividing into smaller tiles and to compute sketches at a higher resolution. We will only need to recurse to depth $\log n$ to go to the resolution of $n$ pixels. Note that we can also amplify the coverage of valid tiles by using a constant number of random shifts of such LSH tilings and in this case we need to recurse into a tile only if it is not fully covered by tiles (from the different shifts) marked as valid. In expectation at least a constant fraction of each side will be covered by some tile that predicts correctly. The same idea works for an intersection of PTFs and also to a union of an intersection of PTFs. For the union of intersection of $k$ PTFs if the boundary curve based on a PTF appears multiple times along the boundary split by intersections we will count them separately for each appearance. This proves the following claim.

**Claim 6.1.** W.h.p. a recursive LSH sketch of depth $O(\log n)$ and size $O(\log n)$ will be able to reconstruct the rectangle. W.h.p. a recursive LSH sketch of depth $O(\log n)$ and size $O(dk \log n)$ will be able to reconstruct an intersection (or union of intersection) of $k$ PTFs of degree $d$ each.

We will see how to further simplify these sketches by merging regions that have similar sketches.

**Stable sketches:** We will call a sketching method stable if for a smooth function, by sketching the function in a constant sized interval (or patch) we get the same sketch regardless of the interval (patch) position. By modifying the $A$ matrix (for e.g. based on the polynomial kernel) we can get stable sketches (see Appendix).

**Hashing sketches to merge duplicates:** We can compress the LSH sketch by merging all near duplicate copies of the sketches from different parts of the same curve – this can be done by further hashing the sketches into another secondary LSH table of sketches and storing only one copy there and keeping only pointers to that table. This can also be done across different images; so the same curve being repeated across images gets replaced by a pointer. This merging can also be done recursively bottom up in a recursive fashion where each patch sketch at a leaf level is replaced by a pointer into the secondary LSH table and going up recursively an entire subtree may be replaced by a pointer so that all copies of a subtree point to the same entry in the global LSH table of sketches. Thus this global LSH table becomes a 'dictionary of sketches of shapes'. This gives us the following.

**Claim 6.2** (Reconstruction). By using stable sketches and merging similar sketches, w.h.p. the above sketch can be compressed to size $O(dk + k \log n)$ for intersection/union of $k$ PTFs of degree $d$ each.

**Claim 6.3** (Indexing). One can use an LSH table as an index of tile-sketches to find shifted copies of the same shape (specified by an intersection of PTFs). Further for a PTF, only a patch containing a part of the PTF is sufficient to index and lookup the shape.

# 7 Learning shapes as concepts

So far we have been viewing images as maps from pixel-locations to pixel-colors. However the same view can be applied to coefficient vectors or (sketch-vectors that can be viewed as smooth transforms of coefficient vectors) for a given shape type (such as ellipses). For simplicity let us just think of shapes in terms of their coefficient vectors that are $d$-dimensional assuming each image corresponding to a shape can be specified by a PTF of degree $d$. Each shape-type can now be viewed as a region in the coefficient space. For example all ellipses can be easily shown to be representable as an intersection of two PTFs in coefficient space .

We proved that the transform from coefficient to sketch space is smooth and invertible; we will extend this to the following assumption. (Note that for PTFs there is a linear map from the sketch vector to the coefficient vector it maps to.)

**Assumption 7.1.** We will assume that transform from the sketch space to the coefficient space is analytic and all derivatives are bounded by some constant. This immediately implies that any low degree polynomial in the coefficient space can be approximated by a low degree polynomial in the sketch space.

**Assumption 7.2** (Model of Shapes in Real-World). A shape is defined by a region (open/closed body with volume) in the coefficient space. These regions are non-overlapping and have a constant margin separating regions corresponding to any distinct shapes. We will assume each region is an intersection of PTFs.

This means that a constant margin PTF separating two shape-types remain separable by a constant margin PTF in the sketch-space. This gives the following theorem.

**Claim 7.3.** Given two shape-types assuming that there is a (robust) high margin PTF that separates them in coefficient space, a polynomial kernel can be used to separate the two regions with a high margin in sketch space.

**Forming a dictionary of shapes:** In fact, we will show how the recursive LSH algorithm described for classifying pixel-locations can actually be applied to classify coefficient vectors. One difference here is that we only get +vely labeled sketches that correspond to some shape in an image; however this is not an issue as we can still study this set of sketches by treating them as "pixels" at a meta level and handle the absence of -vely labeled pixels by randomly sample points as -ve labels (see Appendix). Even for an unsupervised set of images, each contiguous region in coefficient space can now be treated as a new latent shape label and its boundaries can be specified in a robust manner to precisely characterize the region spanned by the sketches corresponding to that shape. This gives the following claim.

**Claim 7.4.** The recursive LSH sketch algorithm when applied to sketches of images can be used to decompose images of shapes into distinct concepts (regions) in sketch space by learning a union of intersection of PTFs. Even when images are not labelled, we can treat each contiguous region in sketch space as a distinct latent concept and characterize the region robustly as intersection of PTFs.

# 8 Experiments

We demonstrate the smoothness of sketch vectors in the latent parameters used to generate the image in this section. As shown in Lemma 5.5, the infinite sketch is smooth as a function of the PTF parameters. Here we further demonstrate that finite sketch vectors also exhibit similar smoothness as we vary the latent parameters.

In particular, we consider the following three functions that generate ellipses, hyperbolas and rectangles images, respectively: $f_1(x,y) = (x-x_c)^2/a^2 + (y-y_c)^2/b^2 - 1, f_2(x,y) = (x-x_c)^2/a^2 - (y-y_c)^2/b^2 - 1, f_3(x,y) = I_{|x-x_c| \leq a}(x,y) \cdot I_{|y-y_c| \leq b}(x,y)$.[2] We sample $(x,y)$ uniformly in $[0,28] \times [0,28]$ and generate pixel values in $\{0,1\}$ with the above functions. The LSH network is trained to predict the correct pixel values and the trained LSH tables are the sketches for the corresponding images.

To examine the smoothness of the sketches, we apply projection of the sketches to a random direction. We also obtain a "raw image" from the functions by computing the pixel values on a uniform $56 \times 56$ grid in $[0,28] \times [0,28]$, and compare with the random projection of the raw images. Examples of such random projections are shown in Figure 1. In Table 1, we calculate the average absolute discrete derivatives of the projection with respect to the parameter, i.e. $\sum_i |\text{Proj}(v_{i+1}) - \text{Proj}(v_i)| / \sum_i |\theta_{i+1} - \theta_i|$, where $v_i$ is the raw image vector (sketch vector) corresponding to parameter $\theta_i$. The sketch vectors have much smaller average absolute discrete derivatives (in Figure 1, the sketch projections are almost flat, compared to large variations in the image projections), indicating a smooth dependency of the sketch vectors on the PTF parameters in the finite dimensional sketch case.

Table 1: Average absolute discrete derivative for varying function parameters

| Shape | Raw image | Sketch |
|---|---|---|
| ellipse | 12.22 | 2.29 |
| hyperbola | 18.31 | 2.63 |
| rectangle | 10.25 | 1.32 |

We also performed an initial set of experiments on using sketches for image classification. In this experiment, we sampled 10% of the train/test split of the fashion MNIST dataset, and obtained LSH

---

[2] $I_A$ is the indicator function of set $A$.

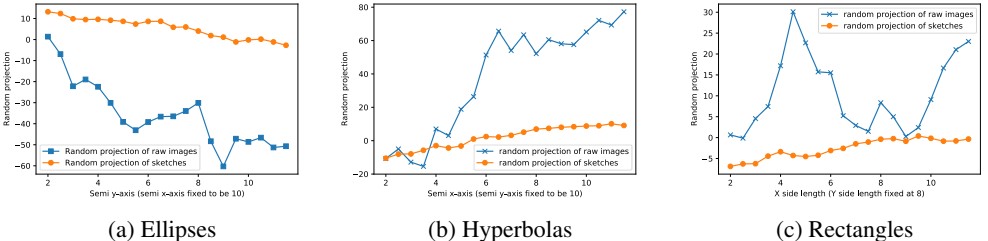

| (a) Ellipses | (b) Hyperbolas | (c) Rectangles |

Figure 1: Examples of 1D random projection of raw images versus sketches. Projections of sketches have smaller variations when we change the parameters in its defining polynomials.

sketches of the images. We then train a 2 layer neural network (with a hidden layer of size 128 and ReLU activation) to classify the sketches. We found a 100% train accuray and 79.3% test accuracy for this task. This demonstrates that sketches have the potential to be used in practical applications.

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
