# A  Additional Preliminaries

**Definition A.1** (Fourier Series)**.** Given a $T$-periodic function $f(x)$, its Fourier series representation is a decomposition into a sum of Fourier basis functions.

$$f(x) = \sum_{k=0}^{\infty} c_k \exp(2\pi i k x / T). \tag{2}$$

The $\{c_k\}_{k=0}^{\infty}$ are referred to as the Fourier coefficients of $x$. For a $(T_1, T_2)$-periodic 2D function $f(x, y)$, its Fourier decomposition is analogously defined as

$$f(x_1, x_2) = \sum_{k,l=0}^{\infty} c_{k,l} \exp(2\pi i k x_1 / T_1) \exp(2\pi i l x_2 / T_2). \tag{3}$$

**Definition A.2** (Discrete Fourier Transform)**.** The discrete Fourier transform transforms a sequence of $N$ complex numbers $\{x_n\}_{n=1}^{N}$ into another sequence $\{\tilde{x}_k\}_{k=1}^{N}$ defined by

$$\tilde{x}_k = \sum_{n=1}^{N} x_n \exp\left(-\frac{i 2\pi k n}{N}\right)$$

Given a function $f : [0,1]^2 \rightarrow [0,1]$ we can define a 1-periodic extension of $f$ to $\mathbb{R}^2$ as follows $f_{periodic}(x_1, x_2) = f(x_1 - \lfloor x_1 \rfloor, x_2 - \lfloor x_2 \rfloor)$. In our work, when we need to consider a Fourier series representation for an image $I$, we use the Fourier series of $I_{periodic}$.

**Definition A.3** (Rademacher Complexity)**.** The Rademacher complexity of a function class $\mathcal{F}$ is a useful quantity to understand how fast function averages for any $f \in \mathcal{F}$ converge to their mean value. Formally, the empirical Rademacher complexity of $\mathcal{F}$ on a sample set $S = (\mathbf{x_1}, \ldots, \mathbf{x_n})$ where each sample $\mathbf{x_i} \sim \mathcal{D}$, is defined as

$$\mathcal{R}_n(\mathcal{F}) = \frac{1}{n} \mathop{\mathbb{E}}_{\boldsymbol{\xi}} \left[ \sup_{f \in \mathcal{F}} \sum_{i=1}^{n} \xi_i f(\mathbf{x_i}) \right],$$

where $\boldsymbol{\xi} = (\xi_1, \ldots, \xi_n)$ is a vector of $n$ i.i.d. Rademacher random variables (each is $+1$ w.p. $1/2$ and $-1$ w.p. $1/2$). The expected Rademacher complexity is then defined as

$$\mathbb{E}[\mathcal{R}_n(\mathcal{F})] = \frac{1}{n} \mathop{\mathbb{E}}_{\boldsymbol{\xi}, \{\mathbf{x_i}\}_{i=1}^{n} \sim \mathcal{D}^n} \left[ \sup_{f \in \mathcal{F}} \sum_{i=1}^{n} \xi_i f(\mathbf{x_i}) \right]$$

Given the above definition of Rademacher complexity, we have the following lemma to bound the worst deviation of the population average from the corresponding sample average over all $f \in \mathcal{F}$ (also known as uniform convergence).

**Lemma A.4** (Theorem 26.5 from [15])**.** *Given a function class $\mathcal{F}$ of functions on inputs $\mathbf{x}$, if for all $f \in \mathcal{F}$, and for all $\mathbf{x}$, $|f(\mathbf{x})| \leq c$, we have with probability $\geq 1 - \delta$,*

$$\mathop{\mathbb{E}}_{\mathbf{x} \sim \mathcal{D}}[f(\mathbf{x})] \leq \mathop{\mathbb{E}}_{n}[f(\mathbf{x})] + 2 \mathbb{E}[\mathcal{R}_n(\mathcal{F})] + c\sqrt{\frac{2\log(2/\delta)}{n}}.$$

**Definition A.5** (Locality Sensitive Hashing (LSH))**.** A locality sensitive hash function is one which maps similar inputs into the same hash bucket. Given a metric space $\mathcal{M} = (M, d)$, an LSH family $\mathcal{F} = \{f : M \rightarrow \mathbb{Z}\}$ mapping points of $M$ to the integers is defined as follows. For any $p, q \in M$ and for any $f$ chosen randomly from $\mathcal{F}$

1. if $d(p, q) \leq R$, then $\Pr[f(p) = f(q)] \geq p_1$,

2. if $d(p, q) > cR$, then $\Pr[f(p) = f(q)] \leq p_2$.

We say the family $\mathcal{F}$ is an $(R, c, p_1, p_2)$-LSH family.

**Definition A.6** (Axis-Aligned Hyperplane LSH for Images)**.** The axis-aligned hyperplane LSH family for 2D images $\mathcal{H}_{\perp}(w_1, w_2)$ is a special case of $\mathcal{H}_{\|\|_2}(k, w)$ where $k = 2$ and the two hyperplanes are chosen to be the $x_1$ and $x_2$ axes respectively. The width parameters for each hyperplane are chosen to be $w_1, w_2$ respectively. Note that we allow for $w_1$ to be different from $w_2$ here.

# B    Full Details on Sketching Low-Complexity Polynomials (Appendix)

Polynomial images are simpler to sketch. For instance, a random projection of the image of a degree-$s$ polynomial to $O(s^2)$ dimensions gives all 4 desiderata, (i) reconstruction, (ii) small sketch, (iii) recoverability of parameters, (iv) smoothness in the polynomial's coefficients. However, it will be beneficial to see how we can use LSH to sketch polynomial images as the analysis will help us extend to more complex images in later sections. In fact, we will show that LSH works not only for sketching degree-bounded polynomials but for polynomials of arbitrary degree as long as their Fourier spectra are appropriately bounded as described in Assumption B.1.

**Assumption B.1.** We assume that the polynomial $p(x_1, x_2)$ generating the image satisfies two properties:

1. Suppose $p(x_1, x_2) = \sum_{k,l=0}^{\infty} c_{k,l} \exp(2\pi i k x_1) \exp(2\pi i l x_2)$ is the Fourier series of $p$. We assume that $\sum_{k,l=0}^{\infty} c_{k,l}^2 (k^2 + 1)(l^2 + 1) \leq \gamma^2$ for some constant $\gamma$.

2. In the region $[0, 1]^2$, the Lipschitz constant of $p(x_1, x_2)$ is bounded by $\Lambda$.

**Lemma B.2** (The Image Norm is Bounded). *We have that*

$$\int_{[0,1]^2} I(x_1, x_2)^2 dx_1 dx_2 \leq \gamma^2.$$

*Proof.* The statement follows from Assumption B.1 and an application of Parseval's theorem. From Parseval's theorem, we have that,

$$\int_{[0,1]^2} I(x_1, x_2)^2 dx_1 dx_2 = \sum_{k,l=0}^{\infty} c_{k,l}^2 \leq \gamma^2. \tag{4}$$

$\square$

We start by studying the limiting behavior of the axis-aligned LSH-Sketch as the number of hash functions $m \to \infty$.

**Definition B.3** (Infinite LSH-Sketch). Given an image $I$, we sample $m$ hash functions from $\mathcal{H}_\perp(m, w_1, w_2)$ for $w_1, w_2 \sim U[1/4, 1/2]$ and we analyze the properties of the resulting sketch as $m \to \infty$.

We will use Fourier analysis for our argument. Since we only have access to a set of finite samples from $I$, we will need the following lemma which says that with a large enough number of samples, the discrete Fourier transform of the sequence of samples is a good approximation to the coefficients of the Fourier series of the continuous function we sampled from.

**Lemma B.4** (Approximation Error of Discrete Fourier Transform (Folklore, for e.g. see [5])). *Given an image $I : [0, 1]^2 \to [0, 1]$ and $n$ randomly sampled pixels from $I$, let $\mathbf{v}$ represent the sequence of pixel values sampled. If $I$ has continuous derivatives and is $\Lambda$-Lipschitz,*

$$|\tilde{I}_n(k) - \tilde{I}(k)| \leq O\left(\frac{\Lambda}{n^2}\right).$$

In this scenario, the metric we focus on for reconstruction is the mean squared error (MSE). Given the LSH Sketch $S(I; m, w_1, w_2)$, let $K = \lceil 1/w_1 \rceil \times \lceil 1/w_2 \rceil$ denote the total number of buckets per hash function. Let $A \in \mathbb{R}^{mK \times n}$ denote the mapping from the $n$ sampled pixel positions $\{(x_1^i, x_2^i)\}$ to buckets. The regularized reconstruction loss we minimize is

$$\|A^\top \mathbf{z} - \mathbf{v}\|_2^2 + \lambda \|\mathbf{z}\|_2^2. \tag{5}$$

Since the above loss resembles the standard ridge regression objective, we can obtain a closed form expression for the $\hat{\mathbf{z}}$ which minimizes the above objective.

**Lemma B.5.** *The sketch which minimizes the reconstruction MSE on train samples is given by*

$$\hat{\mathbf{z}} = A(A^\top A + \lambda I_n)^{-1} \mathbf{v}.$$

*Proof.* Note that since $A^\top A$ is positive semi-definite, for any $\lambda > 0$, $A^\top A + \lambda I_n$ is invertible. The resulting closed form is well-known as the solution for the ridge regression loss. □

Next few Lemmas state some useful properties of the $A$ matrix.

**Lemma B.6.** $\lim_{m\to\infty} A^\top A$ *is a shift-invariant (convolution) matrix which acts also as a smoothening matrix.*

*Proof.* We begin by observing that the $(i, j)^{th}$ entry of $\lim_{m\to\infty} A^\top A$ is proportional the probability that pixels $i, j$ map to same bucket for a random hash function from $\mathcal{H}_\perp(w_1, w_2)$. That is,

$$\lim_{m\to\infty} (A^\top A)[i, j] = \frac{1}{m^2} \Pr_{h\sim\mathcal{H}_\perp(w_1,w_2)}[h(x_1^i, x_2^i) = h(x_1^j, x_2^j)] \tag{6}$$

$$= \frac{1}{m^2} \Pr[E_1] \Pr[E_2], \tag{7}$$

where $E_1$ (r. $E_2$) is the event that $x_1^i, x_2^i$ (r. $x_2^i, x_2^j$) map to same bucket under the first (r. second) hyperplane. Hence we have established that the entries of $\lim_{m\to\infty} A^\top A$ only depend on the distance between the corresponding pixels and hence $\lim_{m\to\infty} A^\top A$ is a shift-invariant (convolution) matrix. In particular the effect of applying $\lim_{m\to\infty} A^\top A$ is similar to that of Gaussian smoothening which leads to attenuation of high frequency components in the input image $I$. To understand the amount of this attenuation we first focus on $E_1$. We observe that $\Pr[E_1] = \max(0, 1 - |x_1^i - x_1^j|/w_1)$. Define the rectangle function

$$\text{rect}(x) = \begin{cases} 1 & \text{if } |x| \leq 0.5 \\ 0 & \text{if otherwise} \end{cases}.$$

If $x_1^i - x_1^j = t$, then $\Pr[E_1]$ is exactly the convolution of $\text{rect}(t/w_1)$ with itself. Therefore the Fourier transform of $\Pr[E_1]$ is the Fourier transform of $\text{rect}(t/w_1)) * \text{rect}(t/w_1)$ which is given by $\tilde{f}(k) = \text{sinc}^2(\pi w_1 k) = \sin^2(\pi w_1 k)/((k^2 + 1)(l^2 + 1))$. Therefore, if we were dealing with 1D inputs, $m^2 \lim_{m\to\infty} A^\top A$ is essentially a convolution of our input function with the triangle function $\max(0, 1 - |t|/w_1)$. In 2D, we get a convolution of our input with the separable function $\max(0, 1 - |t_1|/w_1) \max(0, |t_2|/w_2)$ where $t_1 = x_1^i - x_1^j$ and $t_2 = x_2^i - x_2^j$. Due to separability, this can be viewed as convolving the image with $\max(0, 1 - |t_1|/w_1)$ first and then convolving with $\max(0, |t_2|/w_2)$. Suppose the Fourier transform of the image is denoted by $\tilde{I}(u, v)$. After the application of $\lim_{m\to\infty} A^\top A$, the resulting output's Fourier transform would be given by $\frac{1}{m^2}\tilde{I}(u, v) \text{sinc}^2(\pi w_1 u) \text{sinc}^2(\pi w_2 v)$.

If $w_1, w_2 \sim U[1/3, 2/3]$, then due to linearity of the Fourier transform, the resulting output's Fourier transform would be

$$\frac{1}{m^2} \mathop{\mathbb{E}}_{w_1,w_2} [\tilde{I}(u, v) \text{sinc}^2(\pi w_1 u) \text{sinc}^2(\pi w_2 v)],$$

which is (i) non-zero for all frequencies and crucially (ii) is $\Theta(1/m^2(k^2l^2 + 1))$ for all frequencies large than $k$. □

*Remark* B.7 (Polar co-ordinates). In 2D (or higher dimensions) it is also possible to carry out the Fourier analysis in polar coordinates in a radially symmetric manner via the Hankel transform. This can also be done inside a unit circle instead of $[0, 1]^2$ using spherical harmonics. The analysis becomes more complicated but should produce essentially the same properties.

**Lemma B.8** (Train Reconstruction Error of the Optimal Sketch). *Given $n$ samples, the train reconstruction error of a sketch using $m$ hash functions with $\lambda = O\left(\frac{n}{m}\right)^{1/3}$ is*

$$\|A^\top \hat{z} - v\|_2^2 \leq O\left(\frac{n}{m}\right)^{2/3} \gamma^2,$$

*and the regularization term*

$$\|\hat{z}\|_2^2 \leq O\left(\frac{n}{m}\right)^{1/3} \gamma^2.$$

*Proof.* We have that the train reconstruction error is

$$= \underbrace{\|A^\top \hat{\mathbf{z}} - \mathbf{v}\|_2^2}_{(1)} + \underbrace{\lambda \|\hat{\mathbf{z}}\|_2^2}_{(2)}. \tag{8}$$

Recall that $\hat{\mathbf{z}} = A(A^\top A + \lambda I_n)^{-1}\mathbf{v}$. Let $\lim_{m\to\infty} A^\top A = QDQ^{-1}$ be the eigendecomposition of $\lim_{m\to\infty} A^\top A$. Since $A$ is a convolution matrix, its eigenvectors (columns of $Q$) are the Fourier basis vectors for the DFT over sequences of length $n$. $D$ is a diagonal matrix containing the eigenvalues. We have,

$$(1) = \|(A^\top A(A^\top A + \lambda I_n)^{-1} - I_n)\mathbf{v}\|_2^2 \tag{9}$$

$$= \|(QDQ^{-1}(Q(D + \lambda I_n)Q^{-1})^{-1} - I_n)\mathbf{v}\|_2^2 \tag{10}$$

$$= \|Q(D(D + \lambda I_n)^{-1} - I_n)Q^{-1}\mathbf{v}\|_2^2. \tag{11}$$

Now we express $\mathbf{v} = Q\tilde{\mathbf{v}}$ in the Fourier representation. Then we have,

$$(1) = \|Q(D(D + \lambda I_n)^{-1} - I_n)Q^{-1}Q\tilde{\mathbf{v}}\|_2^2 \tag{12}$$

$$= \|(D(D + \lambda I_n)^{-1} - I_n)\tilde{\mathbf{v}}\|_2^2 \tag{13}$$

Now $D(D + \lambda I_n)^{-1} - I_n$ is a diagonal matrix whose $(i, i)^{th}$ entry is $\frac{\lambda}{\lambda + \lambda_i}$ where $\lambda_i$ is the eigenvalue corresponding to the $i^{th}$ eigenvector of $\lim_{m\to\infty} A^\top A$. From Lemm B.6, the set of $\{\lambda_i\}_{i=1}^n$ will correspond to the set $\{\lambda_{k,l}\}_{(k+1)(l+1)\leq O(n)}$ of the 2D Fourier coefficients of $\lim_{m\to\infty} A^\top A$. We have also shown in Lemma B.6 that $\lambda_{k,l} = \Theta(1/(kl)^2)$. This implies that the diagonal entries of $D(D + \lambda I_n)^{-1} - I_n$ are of the form

$$\frac{\lambda}{\lambda + O(1/(k+1)^2(l+1)^2)}.$$

In addition, we have from Lemma B.4, that the entries of $\mathbf{v}$ correspond approximately to the Fourier coefficients of $p(x_1, x_2)$ up to $\pm O(\Lambda/n^2)$. This implies that

$$\|(D(D + \lambda I_n)^{-1} - I_n)\tilde{\mathbf{v}}\|_2^2 = \sum_{k,l:(k+1)(l+1)\leq O(n)} \left(\frac{\lambda}{\lambda + O(1/(k+1)^2(l+1)^2)}\right)^2 \left(c_{k,l} + \frac{\Lambda}{n^2}\right)^2$$

$$\tag{14}$$

$$\leq \lambda^2 O(\gamma^2). \tag{15}$$

Next, we bound (2). Let $A = USV^\top$ be the SVD of $A$.

$$(2) = \lambda \|A(A^\top A + \lambda I_n)^{-1}\mathbf{v}\|_2^2 \tag{16}$$

$$= \lambda \|USV^\top(VS^\top SV^\top + \lambda I_n)^{-1}\mathbf{v}\|_2^2 \tag{17}$$

$$= \lambda \|USV^\top(V(S^\top S + \lambda I_n)V^\top)^{-1}\mathbf{v}\|_2^2 \tag{18}$$

$$= \lambda \|US(S^\top S + \lambda I_n)^{-1}V^\top\mathbf{v}\|_2^2 \tag{19}$$

$$\leq \lambda \sigma_{\max}^2(S(S^\top S + \lambda I_n)^{-1})\|\mathbf{v}\|_2^2. \tag{20}$$

Now let $\sigma_1, \ldots, \sigma_n$ be the $n$ largest singular values of $A$ (the rest will all be 0). From Lemma B.10, we have that $\sigma_i \leq \sqrt{n/m}$ for all $i$.

$$\sigma_{\max}^2(S(S^\top S + \lambda I_n)^{-1}) = \max_i \left(\frac{\sigma_i}{\sigma_i^2 + \lambda}\right)^2$$

$$\leq \frac{n}{m\lambda^2}$$

The combined train error is then bounded by

$$\lambda^2 O(\gamma^2) + \frac{n}{m\lambda}O(\gamma^2), \tag{21}$$

which is minimized at $\lambda = \left(\frac{n}{m}\right)^{1/3}$. Plugging in this value of $\lambda$ we get total train error is bounded by

$$\leq \left(\frac{n}{m}\right)^{2/3}O(\gamma^2). \tag{22}$$

$\square$

**Lemma B.9.** *The expected test reconstruction error of the optimal sketch is* $O\left(\frac{\gamma}{\sqrt{n}}\right)$.

*Proof.* Since we have a linear problem we can apply we can apply Rademacher complexity based generalization bounds for linear hypothesis classes in our setting. $\square$

**Lemma B.10.** $\|A\|_2 \leq \sqrt{\frac{n}{m}}$.

*Proof.* We have $\|A\|_2 \leq \|A\|_F = \sqrt{\frac{n}{m}}$. $\square$

**Lemma B.11** ($A^\top A + \lambda I_n$ is well-conditioned)**.** $\lim_{m \to \infty} \kappa(A^\top A + \lambda I_n) = 1$.

*Proof.* Let the SVD of $A = USV^\top$. Then,

$$A^\top A + \lambda I_n = VS^\top SV^\top + \lambda I \tag{23}$$
$$= VS^\top SV^\top + \lambda V I_n V^\top \tag{24}$$
$$= V(S^\top S + \lambda I_n)V^\top. \tag{25}$$

Therefore,

$$\kappa(A^\top A + \lambda I_n) = \kappa(S^\top S + \lambda I_n) \leq \frac{\sigma_{\max}^2(A) + \lambda}{\lambda} \tag{26}$$
$$\leq 1 + \frac{n}{m} \cdot \frac{m^{1/3}}{n^{1/3}} = 1 + \left(\frac{n}{m}\right)^{2/3} \tag{27}$$

$\square$

**Claim B.12.** $A$ *has linearly independent columns as* $m \to \infty$.

*Proof.* For any pixel position and a random hash function, the probability that it maps to a bucket where none of the other pixels map to is $> 0$. Hence as $m \to \infty$ the probability of this event happening with at least one hash function goes to 1. This is enough to get linear independence. The only case to take care of is if we get degenerate samples which happens with vanishingly small probability in the first place. $\square$

**Lemma B.13** (Smoothness of the Sketch)**.** *The derivatives of the sketch with respect to the polynomial coefficients which generate our image are continuous and bounded.*

*Proof.* Since $p$ is $\Lambda$-Lipschitz, we have for a specific sample $i$,

$$\|\nabla_{\boldsymbol{\theta}} v_i\|_2 \leq \Lambda. \tag{28}$$

As a function of $\mathbf{v}$, our sketch is given by $\hat{\mathbf{z}} = A(A^\top A + \lambda I_n)^{-1} A\mathbf{v}$. Since $A^\top A + \lambda I_n$ is well-conditioned (Lemma B.11), and $\|A\|_2 \leq \sqrt{n/m}$ (Lemma B.10), this will imply that the derivatives of our sketch with respect to $\mathbf{v}$ are also bounded. By chain rule, we get that the derivatives of our sketch with respect to the polynomial coefficients are also bounded. $\square$

**Lemma B.14** (Recovering the Polynomial's Coefficients from its Sketch)**.** *Given the sketch* $\mathbf{z}$*, there exists a linear transformation* $R$ *such that* $R\mathbf{z} = \boldsymbol{\theta}$.

*Proof.* Recall that $\mathbf{z} = A(A^\top A + \lambda I_n)^{-1}\mathbf{v}$. Here we note that $\operatorname{rank}((A^\top A + \lambda I_n)^{-1}) = \operatorname{rank}(A)$ because $(A^\top A + \lambda I_n)^{-1}$ is a full-rank square matrix and $\lim_{m \to \infty} \operatorname{rank}(A) = n$. With probability 1, we have that the map $\mathbf{v} = G\boldsymbol{\theta}$ also satisfies $rank(G) = p$. This is because $G$ is a Vandermonde matrix which is known to be of full rank as long as none of the points in the sample set coincide. Two points in the sample set coinciding is a measure zero event. This implies that the linear transform from $\boldsymbol{\theta}$ to $\mathbf{z}$ is full-rank with probability 1 implying that it can be inverted to recover $\boldsymbol{\theta}$. $\square$

### B.0.1 Going to a Finite Sized Sketch

So far we have been assuming infinite sized sketch due to infinitely many LSH functions. However we will show how this can be reduced by picking bucket boundaries at a $O(\epsilon)$-grid; that is, the hyperplane shifts are multiples of $\epsilon$ along each axis. For simplicity let us look at the 1-D case and argue that a smooth function can still be approximated using these buckets. In 1-D there are at most $O(1/\epsilon)$ possible distinct buckets. We will only consider the pixels from this $O(\epsilon)$-grid. In this case the Fourier Transform becomes a DFT with $n' = O(1/\epsilon)$ pixels. We already know based on the weighted norm of the image that it can be approximated within error $O(\epsilon)$ using the first $O(1/\epsilon)$ Fourier basis functions which are exactly the DFT basis functions. The above Fourier analysis easily extends to DFT giving a smooth solution with $O(\epsilon)$ error on the $n'$ pixels: the $A$ matrix will be probability of mapping each of these discrete pixels to a bucket with the discrete boundaries; the DFT of the rectangle function also drops in magnitude as $O(1/k)$ for the $k^{th}$ frequency.

For constant $\gamma$, this solution can be extended to the entire domain $[0, 1]$ by predicting the same value for the entire interval between two successive grid points. We will argue that this will introduce at most $O(\epsilon)$ additional error: to see this, note that in the original image the discretization error within an interval depends on the derivative at that grid point. From the $\gamma$-smoothness, We already know the image has a bounded weighted sum of Fourier coefficients which means that the mean squared value of the derivative at these grid-points is also bounded. Thus the discretization error in the original image is at most $O(\epsilon)$. Since the predicted values are within $O(\epsilon)$ average error of the original values at the discretized pixel positions, extending those to the grid intervals will also introduce at most $O(\epsilon)$ error. In general we will need grid length of $\epsilon/\gamma$.

**Claim B.15.** By discretizing the set of pixels to $O(\epsilon/\gamma)$-grid we can get the same guarantees as before by using sketches of size $O(\gamma^2/\epsilon^2)$ in 1-D and $O(\gamma^4/\epsilon^4)$ in 2-D.

## C Full Details about Sketching Thresholded Polynomial Images

Similar to the polynomials setting, we assume that the Fourier spectrum of the underlying polynomial $p$ which we threshold is well-behaved. In addition the previous assumptions we had in Section 4 we also assume an additional condition about the boundary of the PTF. First we present a definition.

**Definition C.1** ($\epsilon$-Boundary of a PTF)**.** The region where the absolute value of the underlying polynomial is $\leq \epsilon$.

The assumptions we make about our PTF image are given next.

**Assumption C.2.**      1. Suppose $p(x_1, x_2) = \sum_{k,l=0}^{\infty} c_{k,l} \exp(2\pi i k x_1) \exp(2\pi i l x_2)$ is the Fourier series of $p$. We assume that $\sum_{k,l=0}^{\infty} c_{k,l}^2 (k^2 + 1)(l^2 + 1) \leq \gamma^2$ for some constant $\gamma$.

   2. In the region $[0, 1]^2$, the Lipschitz constant of $p(x_1, x_2)$ is bounded by $\Lambda$.

   3. The total probability mass of the $\epsilon$-boundary of $p$ is at most $O(\epsilon)$. For example, this holds for an ellipse represented by $ax^2 + by^2 - 1$ when $\sqrt{a^2 + b^2} > c$ for a constant $c$.

We will see how the sketch we construct can accurately reconstruct a PTF image. Due to the binary nature of the true pixel values, an objective such as cross-entropy or the hinge loss is found to be more appropriate for computing the optimal bucket values. In particular, we use the following parameterization of hinge loss. For $y \in \{0, 1\}$, and for $\hat{y} \in \mathbb{R}$,

$$\ell_\epsilon(\hat{y}, y) = \max\left(0, 1 - \frac{\hat{y}y}{\epsilon}\right). \tag{29}$$

Similar to Section 4, we show that the sketch we get here has a low reconstruction error. In particular, we will only make errors near the boundary of the PTF and we can trade off this error by increasing the number of buckets per each hash function. In addition, we show that the manifold containing the sketches is Lipschitz and has bounded derivatives with respect to the coefficients of $p$.

### C.0.1 Properties of Infinite Sketch

**Lemma C.3** (Reconstruction using an Infinite Sketch can generate any Smooth Function over a Finite number of Pixels)**.**

*Proof.* This is mainly because $A^\top$ is going to be full rank with high probability. Firstly, the number of buckets far outnumber the number of pixels in this setting. Next, every Fourier coefficient will be allowed to be non-zero by this transformation. Suppose a sinusoid of a particular frequency $\theta$ maps to 0 when convolved with an LSH of a particular width $w(\theta)$. Then it will necessarily not map to 0 for at least some widths in a small interval around $w(\theta)$. Since we are considering an expected value over randomly drawn widths, this means that at no frequency is the expectation going to be 0. Therefore, we get that reconstruction using an infinite sketch is full rank (equivalent to saying that all Fourier coefficients are free to be set to values we like). Moreover, by bounding the norm of the buckets vector (through regularization), we control the smoothness of the resulting reconstruction. □

**Lemma C.4.** *Constructing an infinite sketch such that each hash function uses constant number of buckets will yield a solution with train reconstruction hinge loss at most $O(\epsilon)$.*

*Proof.* Recall that the hinge loss we optimize is $\max(0, 1 - uv/\epsilon)$. Note that this is 0 when $uv \geq \epsilon$. Given infinitely many hash functions, we can realize a reconstruction such that the reconstructed value at a pixel $u$ is equal to the underlying polynomial value at $v$. This reconstruction will set $|uv^*| > \epsilon$ for $v$ outside the $\epsilon$-boundary and hence incur 0 hinge loss. Therefore, this solution incurs a total loss of $O(\epsilon)$ implying that the optimal solution also incurs a total loss of $O(\epsilon)$. □

**Lemma C.5.** *Given $O(1/\epsilon^2)$ training samples drawn from uniform distribution, we can achieve an average test hinge loss and consequently test $0/1$ error of reconstruction $\epsilon$ via the infinite sketch.*

*Proof.* This is a generalization bound which can be shown by bounding the norm of the values stored in the buckets. This is achieved due to the regularization term. This regularization will imply that $\sum_k c_k^2 O(k^2)$ is bounded where $c_k$ is the $k^{th}$ Fourier coefficient of the reconstruction (before the sign). □

The above two lemmas give us the following theorem about learning PTF images.

**Theorem C.6** (Learning PTFs using Infinite Sketch). *Given $O(1/\epsilon^2)$ uniformly drawn samples from a PTF image, infinite sketch with $O(1)$ buckets per hash function.*

### C.0.2 Smoothness of Sketches

We next prove 5.5.

*Proof of Lemma 5.5.* Let $\mathbf{v}$ be the true pixel values as generated by parameters $\boldsymbol{\theta}$. We will study the case where the sketch is chosen to be the minimizer of the population (aka continuous) loss function. That is, instead of considering a set of $n$ pixels let us look at the setting where we view as the image as a continuous function: $v : [0,1] \times [0,1] \to \mathbb{R}$ with $\mathbf{v}_{\boldsymbol{\theta}}(x) = sign(p_{\boldsymbol{\theta}}(x))$ where $p_{\boldsymbol{\theta}}$ denotes the polynomial with parameters $\boldsymbol{\theta}$. Thus, the sketch is the minimizer of the cost function

$$\mathcal{L}(\boldsymbol{\theta}, \mathbf{z}) = \int_{\mathbf{x}} \ell(\mathbf{v}_{\boldsymbol{\theta}}(\mathbf{x}), \mathbf{u}_{\mathbf{z}}(\mathbf{x})) d\mathbf{x} + \lambda \|\mathbf{z}\|_2^2,$$

where $\mathbf{u}_{\mathbf{z}}(\mathbf{x})$ denotes the predicted image with sketch $\mathbf{z}$ and $\ell$ is the hinge-loss $\ell(a, b) = max(0, 1 - ab/\epsilon)$. The basic idea is that as the sketch is the $\arg\min$ of the above cost function, we can compute its derivatives with respect to the parameters $\boldsymbol{\theta}$ by the *implicit function theorem*. See appendix C.0.2 for the details of the proof.

Abusing notation a little, let

$$\mathbf{z}^*(\boldsymbol{\theta}) = \arg\min_{\mathbf{z}} \mathcal{L}(\boldsymbol{\theta}, \mathbf{z}).$$

As $\mathcal{L}$ is a strongly-convex function, the minimizer $\mathbf{z}^*(\boldsymbol{\theta})$ is also implicitly defined by the equation $\nabla_{\mathbf{z}} \mathcal{L}_{\boldsymbol{\theta}}(\mathbf{z}) = 0$. By using the implicit function theorem, the derivative of $\mathbf{z}^*(\boldsymbol{\theta})$ can be written (cf. [14] for instance) as

$$\nabla_\theta z^*(\theta) = (\nabla_z^2 \mathcal{L}(\boldsymbol{\theta}, z^*))^{-1} \partial_{z^*} \partial_{\boldsymbol{\theta}} \mathcal{L}(\boldsymbol{\theta}, z^*). \tag{30}$$

Note that by the regularizer in $\mathcal{L}$, $\nabla_z^2 \mathcal{L}(\boldsymbol{\theta}, z) \succeq 2\lambda I$ so that $\|(\nabla_z^2 \mathcal{L}(\boldsymbol{\theta}, z^*))^{-1}\| \leq 1/2\lambda = O(poly(1/\epsilon))$. It remains to bound the norm of the second term.

It follows from elementary calculus that

$$\nabla_\theta \mathcal{L}(\theta, z^*) = \int_x \frac{u_{z^*}(x)}{\epsilon} \delta(p_\theta(x)) \nabla_\theta p_\theta(x) dx = \int_{x:p_\theta(x)=0} \frac{u_{z^*}(x)}{\epsilon} \nabla_\theta p_\theta(x) dx, \qquad (31)$$

where $\delta(\ )$ denotes the Dirac delta function.

Now, recall that $u_z(x) = \sum_{j=1}^m z_j A_j(x)$ where $A_j(x) = 1$ if $x$ is hashed to bucket $j$ and 0 else. Therefore,

$$\partial_{z_j} \nabla_\theta \mathcal{L}(\theta, z^*) = \int_{x:p_\theta(x)=0} \frac{A_j(x)}{\epsilon} dx.$$

Therefore, the norm of the vector $\partial_{z^*} \partial_\theta \mathcal{L}(\theta, z^*)$ is bounded.

A similar argument with more detailed calculations shows that even higher order derivatives of $z^*(\theta)$ with respect to $\theta$ exist and are bounded (we start with identities in Equations 30, 31, and use convenient identities such as $\int_x \delta'(x)\phi(x)dx = -\int_x \delta(x)\phi'(x)dx = \phi'(0)$).

In contrast, we next argue that the mapping $\boldsymbol{\theta} \to \mathbf{v_\theta}$ while continuous is not necessarily differentiable in $\boldsymbol{\theta}$. To see this, consider a simple one-dimensional scenario where the PTF itself is of degree 1 and corresponds to the simple step function with one underlying parameter $t \in \mathbb{R}$ the step threshold. We are interested in the mapping that maps $t$ to the corresponding step function $f_t(x) = 1$ if $x \geq t$ and 0 otherwise. Viewing this as a mapping from $\mathbb{R}$ to the Banach space of square-norm bounded functions on $\mathbb{R}$, we have $\lim_{h \to 0}(f_{t+h} - f_t)/h$ is the Dirac delta function centered at $t$ which technically is not bounded in 2-norm. Note that this argument essentially implies that if we take images of rectangles and slide one of the boundaries, random projections of these images is a Brownian motion. Since the space covered by a Brownian motion is highly non-smooth we can conclude that the region formed by such images is also non-smooth $\qquad \square$

# D  Stable sketches

Our Fourier analysis shows that LSH sketch acts like a Fourier basis kernel where the $jth$ frequency gets scaled by $1/j$. Further a somewhat different $A$ matrix can be used to tune the level of smoothness scale down higher frequencies at different levels and can even be used to essentially restrict the reconstructed image to the first $d$ Fourier frequencies. Instead of using the Fourier basis functions We could also use the polynomial kernel corresponding to the first $O(d)$ monomials then we can exactly get the polynomial coefficients representing the function. Note that we only need to know some upper bound on the degree of the polynomial.

**Claim D.1.** By using a polynomial kernel one can learn a degree $d$ polynomial using an $O(d)$ sized sketch. Further the sketch can be obtained from any patch of constant size. This gives us a stable sketch; that is, the same sketch is obtained regardless of which patch it is computed on: precisely, given a $d$ polynomial of degree at most $d$ and its evaluations at pixels in an interval of constant length $[a, b]$ within error $\epsilon = exp(-\Omega(d \log d))$ we can determine its coefficients within error $\epsilon$.

*Proof.* The main point is even if we pick $d$ points at gaps of $1/d$ then the vandermonde matrix has condition number at most $exp(d \log d)$. $\qquad \square$

Instead of using a polynomial kernel one can also a more natural hash matrix $A$ that is inspired from the normal distribution. The main point is that the Fourier transform of a normal distribution of a high width is also a normal distribution of low width which becomes like a low pass filter. We can use a discretized version such a matrix to exactly have non-zero weights only on the first few frequencies.

**Claim D.2.** In 1-dimension with $2n' + 1$ equispaced pixels denoted by $x$ in the range $[-n'..n']$ the DFT of the function $\cos^{2l}(\pi x/(2n'))$ is a low pass filter with non-zero values only for frequencies in the range $[-3l, 3l]$. We can set $n' = O(1/\epsilon)$ to get an $A$ matrix $A(x, y) = \cos^{2l}(\pi(x - y)/(2n'))$ that maps the pixels to $O(1/\epsilon)$ buckets so that $A^\top A$ functions precisely as a low pass filter that zeros out frequencies higher than $O(l)$.

*Proof.* Since the DFT and its inverse are symmetric, look at the discrete function $f(t)$ on $2n' + 1$ pixels in the range $[-n'..n']$ where $f(0) = 1/2, f(-1) = f(+1) = 1/4$ and $f$ is 0 at all other

points. In the frequency domain it can be written as $F(k) = e^{jk\pi/n'}/4 + e^{-jk\pi/n'}/4 + 1/2 = (\cos(k\pi/n') + 1)/2 = \cos^2(k\pi/(2n'))$

Now if we take the self convolution of $f$ $l$ times its DFT will be $F^l = \cos^{2l}(k\pi/(2n'))$. Further since $f$ is non-zero only in the range $[-1, 1]$ by a simple inductive argument, $f$ convoluted $l$ times is non-zero only in the range $[-3l, 3l]$. And all its values in that range are at least $1/4^l$ $\qquad\square$

The argument for learning stable sketches can also be applied to Taylor series. If the curve can be written as $y = f(x)$ where $f$ is an analytic function, then one can learn the initial few coefficients of the Taylor series expansion of $f$ that approximate $f$ within a tiny error (if the curve takes multiple values at a given $x$ we can break it into regions and use random rotations so that this is not the case in at least one rotation). Again these set of coefficients form a sketch that is independent of which patch it has been computed from as long as the patch has a significant part of the curve. If there are $n$ pixels one can approximate the first $\Omega(\log n)$ coefficients. Thus the above result can be extended to analytic functions with bounded derivatives.

**Claim D.3.** Given an analytic function that converges in $[0, 1]$ with all derivatives bounded in magnitude by some constant in the range $[0, 1]$ and its evaluations at points in the range $[a, b]$ within error $\epsilon$ we can determine its value in the entire range within error $\epsilon^{O(1)}$. A sketch of this extrapolation can serve as a stable sketch of the function.

*Proof.* If is well known that if we take say $d$ points $x_i, ., x_d$ for an analytic function $f(x)$ taking values $y_1, .., y_d$ then the polynomial $f(x) = \sum y_i \Pi_{j\neq i}(x - x_j)/\Pi_{j\neq i}(x_i - x_j)$ fitted on these $d$ points has error at most $f^{(d+1)}(\eta)/(d+1)!\Pi(x - x_i)$ in the entire interval $[0, 1]$ for some $\eta \in [0, 1]$. Since all derivatives are constant this is at most $1/d! = exp(-O(d\log d)) = \delta$ (say). There is also the contribution from error in the measurements $y_i$ in the extrapolation formula $f(x) = \sum y_i \Pi_{j\neq i}(x - x_j)/\Pi_{j\neq i}(x_i - x_j)$. For the error in this to be less than $\delta$ we need that the measurements $y_i$ need to be such that $y_i \Pi_{j\neq i}(x - x_j)/\Pi_{j\neq i}(x_i - x_j)$ has error at most $\delta/d$. If we choose all $x_i$ in some constant sized interval to be values at least $O(1/d)$ apart then it means that the we want $y_i$ to be accurate up to $\delta O(1/d)^d = exp(-O(d\log d)) = \delta^{O(1)}$. Thus if the error in the measurements is at most $\epsilon$ then we can extrapolate within error $\epsilon^{O(1)}$ in the entire interval $[0, 1]$. This becomes the stable sketch for $f$. The value of $d = O(\log(1/\epsilon)/\log\log(1/\epsilon))$ $\qquad\square$

The Taylor series approach can be used even for PTFs by fitting the points on the boundary to a polynomial.

# E    Intersection of PTFs

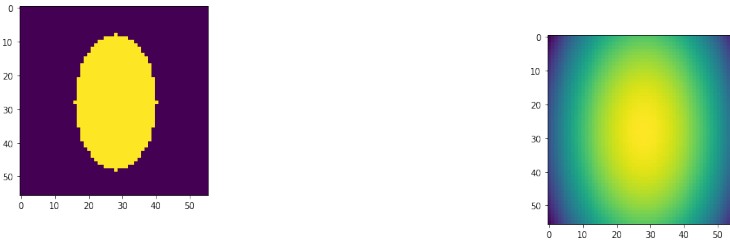

Figure 2: Plot of a thresholded ellipse (PTF) and an ellipse as a polynomial

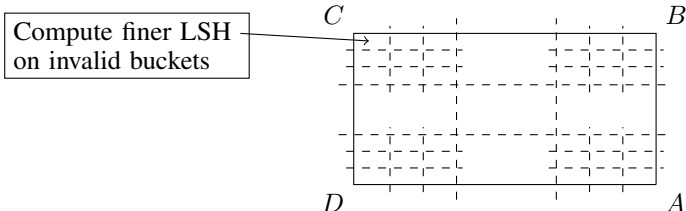

Figure 3: Rectangle tesselation based on Recursive Sketching Subroutine (Algorithm 1)

*Remark* E.1. [Tile sketches can be viewed as a smoother function]

The sketches can be viewed as a map from tile-positions to tile-sketches; this itself can be viewed as an "image mapping pixels (tile positions) to $d$-dimensional sketch vector (instead of value in $R$ its a value in $R^d$). Note that this map can be viewed as a smoother than the original map from pixels to color as for most tiles of size $1/c$ their sketch will be the same if the tile is shifted by $O(1/c)$. The values stored at such tiles is not affected by a smoothening of the function by $O(1/c)$. Such tiles are sufficient to reconstruct the contents of other tiles that are covered by such tiles. This smoothend version of the map can be stored at a lower resolution to capture the image and hence methods described earlier can be applied to get a sketch of the map. Thus If we use a few random shifts of tilings and take a smoothened version of this map from tile-positions to tile-sketches, based on remark 4.5 we get a representation that is invariant to the random shift choices.

If we use these the sketches of the edges as features, then on top of these we will get a simple classifier for the rectangle. Note that a union of intersections over $k$ features can be written as a degree $k$ polynomial over the features which can be easily learned for small $k$.

**Claim E.2.** By LSH hashing the stable sketches from the tiles and using the top $k$ sketches as features, we can build a simple classifier of size $dk$ for a union of intersection of $k$ PTFs of degree $d$.

**LSH table of sketches as an inverse map:** Note also that the LSH table of sketches can be viewed as a map from a patch-sketch to its tile-coordinate. This can be easily used to identify shifted copies of an image. For two shifted images the outputs of their maps will differ exactly by the shift value. One point to note is that a single patch bucket may correspond to multiple tiles – for such patches one would need to store a "set" of positions per bucket. It is possible to store a sketch of a set so that two shifted sets will have just shifted sketches. This can be done by first shifting the sets to make them mean zero and then sketching the sets; The set can again be viewed as a map from tile-positions to 0/1 depending on whether they are in set and then sketching this map based on Remark E.1.

*Remark* E.3 (Relation to Vision Transformers). Note that after merging tiles corresponding to similar sketches, one can learn a single sketch for the entire merged set of tiles. This is reminiscent of image transformer where related patches can attend to each other and merge into one token.

*Remark* E.4. [Shift resistant duplicate representation] Given two identical images that are shifts of each other they can be made "shift free" by first centering the image by say shifting the centroid to the origin. This will produce the same recursive sketch that can be stored once in the LSH table of sketches – this would be despite getting different random samples of pixels in the two images as per remark 4.5. This can even be applied to parts of an image. For example think about the same (say top-left) corner in two different (axis-aligned) rectangles. After the first level of LSH bucketing is done we get the tiles marked invalid that contain the corner. We can center the region covered by the union of these invalid tiles and then recursively continue the sketching. If we used sufficiently many shifts, the same corner from two different rectangles in different images will produce in expectation the same region after centering. Thus all say "top-left" corners of axis aligned rectangles will get the same recursive sketches in expectation.

So far there are several different methods that we have described for computing sketches: LSH sketch, polynomial kernel, Taylor series interpolation, the foward and the inverse maps from tile-positions to tile-sketches, mean and moments for a set. Different methods provide different guarantees. By keeping a "heterogenerous combination" of all these different sketches we get a single combined sketches different parts of which can be used for reconstruction, locality-properties, indexing etc.

**LSH table of sketches as an inverse map:** Note that the LSH table of sketches can be viewed as a map from a patch-sketch to its tile-coordinate. This can be easily used to identify shifted copies of an

image. For two shifted images the outputs of their maps will differ exactly by the shift value. One point to note is that a single patch bucket may correspond to multiple tiles – for such patches one would need to store a "set" of positions per bucket. We can also store a sketch of a set so that two shifted sets will have just shifted sketches. This can be done by first shifting the sets to make them mean zero and then sketching the sets; the set can again be viewed as a map from tile-positions to 0/1.

Note that since are sketch is a recursive sketch of tiles, it has the local property in the main proposition as the sketch of a part of an image easily be obtained form the full sketch.

**Claim E.5** (Indexing)**.** One can use an LSH table as an index of tile-sketches to find shifted copies of the same shape (specified by an intersection of PTFs). Further for a PTF, only a patch containing a part of the PTF is sufficient to index and lookup the shape.

*Remark* E.6. [Intersections of PTFs in higher dimensions] For the intersection of PTFs we assumed so far 2-d images. There are some complexities that arise when we go to higher dimensions. For example if we look at the intersection of two halfspaces, if we use a recursive LSH, the edge formed from the intersection produces several small tiles (3-d tiles). If there are $n$ grid points (pixels) total in 3-d then the edge will produce an $O(n^{1/3})$ sketch due to the tiles along the edge at the leaf level of the recursion; in $k$-dimensions there will be $O(n^{1-1/k})$ tiles. This can be avoided by clustering all such tiles at each depth of the recursion and learning a single smooth function for the entire union of those tiles at the next level of granularity that approximates the edge. Note that there is low degree polynomial (for example the equation of a cylinder along the edge) that is like a smoothened version of the edge. By decreasing the radius of the cylinder in each depth of the recursion we get better approximation to the edge each time. This gives an $O(\log n)$ sized sketch for each edge in a 3-d shape.

Note that even though we are using stable sketches for the above claims, even we if did not use such stable sketches and sketches of the same curve differ slightly across tiles we can look at similar sketches in adjacent tiles and try to merge them by finding if there is a single sketch that explains the union of adjacent tiles.

# F   Learning shapes as concepts

**Claim F.1.** Ellipses satisfy Assumption 7.2.

*Proof.* Look at the coefficients of the first six monomials in the expansion: $x^2, y^2, xy, x, y, 1$. Let the coefficients of the first three terms be $a, b, c$ respectively. The coefficients will in fact define an ellipse if the corresponding $2 \times 2$ matrix $[[a c/2][c/2 b]]$ is positive semi-definite. This in turn is equivalent to three conditions: $a > 0$, $b > 0$ and $4ab > c^2$. Thus, in the coefficient space, ellipses correspond to an intersection of three PTFs. Look at the coefficients of the first six monomials in the expansion: $x^2, y^2, xy, x, y, 1$. Let the coefficients of the first three terms be $a, b, c$ respectively. The coefficients will in fact define an ellipse if the corresponding $2 \times 2$ matrix $[[ac/2][c/2b]]$ is positive semi-definite. This in turn is eqauivalent to three conditions: $a > 0$, $b > 0$ and $4ab > c^2$. Thus, in the coefficient space, ellipses correspond to an intersection of three PTFs. $\square$

*Remark* F.2. [Sketching with only positively labeled pixels] Our sketching methods work even if do not have any negatively labeled pixels and only have pixels from inside the (union) of PTF. One difference is that we only get positively labeled points, that is, points inside the shape-region. We will show how negative labeled points can be generated by random sampling pixels from the space of pixels (say in $R^k$). Although this will introduce noise within the positive region this can be corrected by correcting each label to be positive if in some positive label in an $\epsilon$-neighborhood around it. If we sample $\tilde{\Omega}((\sqrt{k}/\epsilon)^k)$ points in an $\epsilon/\sqrt{k}$-grid as negative labels then after correction all points will be correctly labeled except for points within $\epsilon$ of the boundary of a PTF.

We will assume that each shape type forms a disjoint region in coefficient space. And that union of these regions can be specified as a union of intersection of PTFs. Thus we can apply the algorithm from previous section (see remark E.6). A contiguous region of tiles can be identified as a separate shape/concept.