# OpenReview forum: "Sketching based Representations for Robust Image Classification with Provable Guarantees"
_NeurIPS.cc/2022/Conference — NeurIPS 2022 Accept_

### Official Review · Reviewer_kMv5 · 2022-07-10

**Rating:** 5
**Confidence:** 1
**Soundness:** 3 good
**Presentation:** 2 fair
**Contribution:** 2 fair

**Summary:**

This paper studied the question "Do deep networks learn representations that are isomorphic to the parameters to generate the image?". It provided a sketching algorithm to produce an image sketch that is equivalent to the parameters. The main idea is to use a locality sensitive hash (LSH) table. The authors provided theoretical results on synthetic images that are composed of a union or intersection of several mathematically specified shapes using thresholded polynomial functions (for e.g. ellipses, rectangles).

**Questions:**

Do the theory proposed in this paper have potential applications on real classification tasks?

**Limitations:**

Addressed by the authors.

**Strengths And Weaknesses:**

Strengths:
The methods seems to have theoretical supports. The results seems to be solid.

Weaknesses:
It lacks intuitive explanations behind the theoretical results. Also, it does not tell the potential usage of the proposed method in real applications.

---

> ### Author Response · Authors · 2022-08-02
> **Author's response to the questions**
>
> Thanks for your review and the feedback provided. We are glad that you found our theoretical results appealing. We try to address your concerns and questions below.
>
> 1. Intuitive explanations behind the theoretical results: Thanks, we have added visual images to show the sketching in increasing cases of complexity such as simple ellipses, PTFs and rectangles to serve as an intuitive explanation.
>
> 2. Potential usage of proposed method:
>  * Our sketching framework is much more broadly applicable beyond the 2D images setting we presented in the paper. Our method is generally applicable to represent any function and not just 2D images. Even speech can be viewed as a 1D function. Videos can be viewed as 3D functions. Higher level representations can be viewed as functions themselves.
> * The main essence of our approach is about extracting some simple structure that captures the essential information of these objects (be they images, videos, speech signals or text). In that sense, we view our approach as having potential as a general machine learning approach that is alternative to the current deep learning paradigm. We will emphasize this discussion in the main body in future versions.
> * We also performed an initial set of experiments on using sketches for image classification. In this experiment, we sampled 10% of the train/test split of the fashion MNIST dataset, and obtained LSH sketches of the images. We then train a 2 layer neural network (with a hidden layer of size 128 and ReLU activation) to classify the sketches. We found a 100% train accuray and 79.3% test accuracy for this task. This demonstrates that sketches have the potential to be used in practical applications. We will update the paper with more detailed study on this experiment.

---

### Official Review · Reviewer_JF41 · 2022-07-11

**Rating:** 6
**Confidence:** 3
**Soundness:** 3 good
**Presentation:** 4 excellent
**Contribution:** 3 good

**Summary:**

This work aims to describe the sketch of a simple image using polynomial functions and hash tables. This is a handcrafted feature technique that has interpretation compared to deep learning.

**Questions:**

I hope the authors will add a large number of experiments to prove its value as a feature.

**Strengths And Weaknesses:**

Overall, I really like the idea of this paper, especially in this era of deep learning run amok.
Not all applications have sufficient samples, and handcrafted features become especially important.

Strengths

[+]  The motivation sounds reasonable and the writing of the paper is good.

[+]  The methodology is also reasonable, considering the trade-off between read efficiency (Hash) and accuracy (Polynomial).

Weaknesses

[-]  My only concern is the experimental part, which is difficult to persuade me that as a feature should be tested by the classification task.

---

> ### Author Response · Authors · 2022-08-02
> **Author's response to your questions**
>
> Thanks for your review and for the feedback provided. We are glad that you found our motivation and methodology appealing. We hope to address some of your concerns about experiments and other questions below.
>
> 1. Experiments to support the efficacy of the proposed method: we performed an initial set of experiments on using sketches for image classification. In this experiment, we sampled 10% of the train/test split of the fashion MNIST dataset, and obtained LSH sketches of the images. We then train a 2 layer neural network (with a hidden layer of size 128 and ReLU activation) to classify the sketches. We found a 100% train accuray and 79.3% test accuracy for this task. This demonstrates that sketches have the potential to be used in practical applications. We will update the paper with more detailed study on this experiment.

---

### Official Review · Reviewer_Jdpa · 2022-07-11

**Rating:** 5
**Confidence:** 2
**Soundness:** 3 good
**Presentation:** 3 good
**Contribution:** 3 good

**Summary:**

The scope of this work falls into image representation learning, and the authors aim to study synthetic images with thresholded polynomial functions (in a generative process). In particular, the authors proposed a sketching algorithm by using locality sensitive hash (LSH). As a result, a succinct sketch is produced from an image, such that the sketch can be used for image reconstruction. The major contribution is that the authors proposed to leverage LSH tables to split an image into regions, then train parameters within each hash bucket, and the resulting parameters can be used to reconstruct image regions. The authors claim potential usage on sketches for image classification and possible extension of using secondary LSH index of sketches for different parts/shapes in the image as well.

**Questions:**

I think the authors could provide some possible applications of using the proposed sketching method, where image classification could be one with a very broad impact on the computer vision community.

In addition, is that possible to some visual examples of handling simple and complex images by the proposed method, so that readers can more quickly get the point of what and how the algorithm can do.

**Ethics Review Area:**

["I don’t know"]

**Limitations:**

The authors only study synthetic images with basic shapes, while there might be issues when dealing with more complex images or real images.

**Strengths And Weaknesses:**

Strengths

The overall idea makes sense to me, and the authors provide detailed descriptions of the method, including problem setup, preliminaries, and many additional explanations of the proposed definitions, lemma so on, making the read more smooth (although I have to repeatedly read several times). More details are also provided in the supplementary material.

Weaknesses

The experiments part is somehow limited. The authors only demonstrate the smoothness of the sketch vectors on three functions that can generate simple shapes (ellipses, hyperbolas and rectangles). However, there is no support for the main claim about the usefulness of image classification in the paper title.

---

> ### Author Response · Authors · 2022-08-02
> **Author's response to your questions**
>
> Thanks for your review and the feedback provided. We address the main concerns and questions below.
>
> 1. Applications of the Proposed Method: Our sketching framework is much more broadly applicable beyond the 2D images setting we presented in the paper. Our method is generally applicable to represent any function and not just 2D images. Even speech can be viewed as a 1D function. Videos can be viewed as 3D functions. Higher level representations can be viewed as functions themselves.
> The main essence of our approach is about extracting some simple structure that captures the essential information of these objects (be they images, videos, speech signals or text). In that sense, we view our approach as having potential as a general machine learning approach that is alternative to the current deep learning paradigm. We will emphasize this discussion in the main body in future versions.
> We performed an initial set of experiments on using sketches for image classification. In this experiment, we sampled 10% of the train/test split of the fashion MNIST dataset, and obtained LSH sketches of the images. We then train a 2 layer neural network (with a hidden layer of size 128 and ReLU activation) to classify the sketches. We found a 100% train accuray and 79.3% test accuracy for this task. This demonstrates that sketches have the potential to be used in practical applications. We will update the paper with more detailed study on this experiment.
>
> 2. Visual Examples: Thanks, we have added visual images to show the sketching in increasing cases of complexity such as simple ellipses, PTFs and rectangles to serve as an intuitive explanation.
>
> 3. Issues when dealing with more complex shapes: We view synthetic shapes such as ellipses or low-degree polynomials as an important basis step to achieve larger goals such as general shape recognition. Since we aim to show that our method works theoretically, we do have to assume some sort of a restriction on the shapes we see in our images but in practice, we believe our method can generalize beyond these simple shapes.

---

### Meta-Review · Area_Chair_kUB7 · 2022-08-25

**Recommendation:** Accept
**Confidence:** Certain

**Metareview:**

All reviewers are positive about this paper. After rebuttal, the authors have well solved the reviewers' concerns, and  improve the quality of this paper. So I suggest accepting this paper.

**Award:**

No

---

### Decision · Program_Chairs · 2022-09-14

Accept